# Programmed Cell Death Reversal: Polyamines, Effectors of the U-Turn from the Program of Death in *Helianthus tuberosus* L.

**DOI:** 10.3390/ijms25105386

**Published:** 2024-05-15

**Authors:** Donatella Serafini-Fracassini, Stefano Del Duca

**Affiliations:** 1Department of Biological, Geological and Environmental Sciences, University of Bologna, 40126 Bologna, Italy; donatella.serafini@unibo.it; 2Interdepartmental Centre for Agri-Food Industrial Research, University of Bologna, 40126 Bologna, Italy

**Keywords:** *Helianthus tuberosus* L., programmed cell death, amitotic cell cycle, dormancy, polyamines, putrescine, spermidine, spermine, transglutaminase, tuber, differentiation, abiotic stress, growth

## Abstract

This review describes a 50-year-long research study on the characteristics of *Helianthus tuberosus* L. tuber dormancy, its natural release and programmed cell death (PCD), as well as on the ability to change the PCD so as to return the tuber to a life program. The experimentation on the tuber over the years is due to its particular properties of being naturally deficient in polyamines (PAs) during dormancy and of immediately reacting to transplants by growing and synthesizing PAs. This review summarizes the research conducted in a unicum body. As in nature, the tuber tissue has to furnish its storage substances to grow vegetative buds, whereby its destiny is PCD. The review’s main objective concerns data on PCD, the link with free and conjugated PAs and their capacity to switch the destiny of the tuber from a program of death to one of new life. PCD reversibility is an important biological challenge that is verified here but not reported in other experimental models. Important aspects of PA features are their capacity to change the cell functions from storage to meristematic ones and their involvement in amitosis and differentiation. Other roles reported here have also been confirmed in other plants. PAs exert multiple diverse roles, suggesting that they are not simply growth substances, as also further described in other plants.

## 1. Introduction

### Development and PCD

The development of a plant is very complex and is coordinated among its different organs, even if they are very far from each other. A system of transport consisting of either internal or external signals must be involved. Many factors regulate the developmental events that have, as the final target, the reproduction of the organism. During differentiation, parallel to the development of a new group of cells, the elimination of other cells also occurs. The last process, occurring with different morpho-functional modalities, can be defined under the term “programmed cell death” (PCD) or, better, DCD (developmental cell death). This is a process that occurs in every plant throughout its life span and constitutes physiological steps during the entire process. It is a process driven by an internal programmed control of development, which has the precise target to maintain plant homeostasis [1,2,3].

The latter requires the selective and programmed elimination of groups of cells of entire organs (i.e., leaves, petals, tubers, stamina, fruits, pollen, etc.) [4,5,6,7,8,9,10,11,12] or even of the entire organism that are no longer necessary (as, for example, monocarpic plants after seed production). In some cases, PCD can be induced by a lot of occasional external factors, like environmental stresses and pathogen infections [13,14]. PCD events can be regarded as groups of coordinated successive processes, which at the end block all the fundamental cell metabolisms required for life, causing cell death, whereby nutrients are re-used by other cells. The process proceeds in an orderly and predictable manner, irreversibly programmed until death. The irreversibility of PCD is an important biological challenge to be verified and is performed by examining the peculiar example of a dying organ.

Different examples of the developmental type of PCD in various plant organs have recently been summarized [2], underlining the diffusion of this event. Xie and collaborators [15] describe how diffuse the PCD is in different multiple flowering structures. PAs have a regulative role in reproductive organs, like in petals or in the response to self-incompatible pollen [6,7,10,16], but also in senescent leaves either excised or not [12] and in the hypersensitive response, as well as in the tuber parenchyma during dormancy release [17]. In all the models described above, the molecular mechanism appears similar to that described during apoptosis in animal cells [18].

The plant *Helianthus tuberosus* L. (*Helianthus t*.), cultivated since 1964 in the Botanical Garden of Bologna University and vegetatively reproduced until recently, was chosen for the peculiar characteristics of its tuber, which, in nature, during its natural release from dormancy, undergoes programmed cell death (PCD) [17]. This type of irreversible PCD can be experimentally reverted by explanting and culturing in vitro its dormant parenchyma in the presence of polyamines (PAs), giving rise to a new life program. In many PCD organisms, among other complex morpho-functional factors which control it at the transcriptional, translational and post-translational levels, PAs are now recognized to exert important roles. These biogenic amines are ubiquitous across almost all living cells, from bacteria to plant and animal phyla, and are essential for life. In plants, the regulatory role in PCD of the main three aliphatic PAs, i.e., putrescine (Put), spermidine (Spd) and spermine (Spm), is reported by several articles [8,9,15,16,17,18,19,20,21,22].

The *Helianthus t.* experimental system allows us to clarify how these cells can degrade a part of their primary metabolites, changing their program from one destined for death to instead develop a new life program. Through this system, it has been possible to open another important field of plant research, furnishing the milestones of PA knowledge: PA roles were detected for the first time in plants.

In the early sixties, PAs were practically unknown in plants, having been detected only in seeds, where a role during growth had only been hypothesized [23]. The present review reports the numerous morpho-functional data on *Helianthus t.* obtained since the beginning of this research. The research continued over the years until recently, and expanded from the data obtained by the Bologna laboratory personnel to a widespread number of papers on many other plants in other laboratories.

This review covers data reported on the literature starting from the sixties; the most significant articles of the period 1960–1990 have been cited. Naturally, the review also considers the subsequent period up to today, demonstrating the importance of polyamines for modern-day research.

As many PA characteristics are mentioned very frequently throughout this text and since it is impossible to describe them in detail each time, a separate Appendix A, provided with figures (Appendix A), is added to ensure that the many explications herein do not interrupt the text; this part is specifically addressed to the readers not acquainted with these substances to furnish essential data. Moreover, the following second part aims to evidence the milestones attained by this research to present a sort of history of this long research.

## 2. The *Helianthus tuberosus* Experimental System

The *Helianthus t.* (Jerusalem artichoke, topinambur, wild sunflower, kaentawan (Tai), kiku-imu (Japan), pataca (Spain)) shrub (Figure 1A–C) is an angiosperm of the Asteraceae family, originating from North America. Its multiple names show that it is well known in the world. The name Jerusalem artichoke has nothing to do with Jerusalem but derives from the incorrect pronunciation from the Italian immigrants in the USA of the term “girasole” (sunflower), having a similar flower. “Artichoke” is due to the similarity of their taste with the former.

The plant of *Helianthus t.* cv OB1 utilized in this study was generously gifted by Prof. Roger J. Gautheret, who selected it and pointed out already in 1953 at Paris University the technique of in vitro culture of this line for its remarkable properties. Notably, this line remains exceptionally stable and does not develop roots in culture, as described below. Since 1964, the cv. OB1 line has been vegetatively propagated and cultivated in the Botanical Garden of the Department of Biological, Geological and Environmental Sciences at the University of Bologna. Dr Umberto Mossetti, the current director, oversees this garden. During the 1980s, Dr. Mossetti, as a scholarship recipient, published some articles on tuber dormancy and in vitro culture, including the first data on bound PAs [25,26,27].

The plant stem with flower buds (Figure 1A), the developed dormant tubers showing the primordia of vegetative buds (Figure 1B) and the mature flower (Figure 1C) are shown. This plant is characterized by the formation, after flowering, of ephemeral subterraneous tubers which act as storage organs, filled up by many substances (Figure 1B).

Figure 2 outlines the plant’s developmental stages throughout the year, from shoot formation to tuber sprouting in the temperate climate of the northern hemisphere.

The dormant tubers are used for many morphological and functional studies and even widely consumed as food [28]. For their low glucose and lipid content (starch is almost absent), they are used for diets, especially after chemotherapy and for diabetic patients; moreover, they are utilized for biofuel, feed, bioactive compounds as antioxidants, cosmetics, etc. As examples, see the reviews [28,29,30] which describe the chemical, amino-acid, vitamin and mineral composition of tubers and the pharmacological, therapeutic, nutrient, toxic and prophylactic effects on the human organism.

In autumn, the aerial stem dies, whereas the tubers survive a period of suspended growth during winter, being deprived of growth substances. After mid-winter, tubers release dormancy and begin to transfer the storage to their buds which then germinate, accomplishing vegetative reproduction and thus allowing the survival of the species. In the meantime, tuber cells begin to die by innate PCD: in natural conditions, cells are regulated by external environmental factors, for example, temperature changes. Alternatively, the tubers conserved during winter in an artificial environment under wet sand at 4 °C cannot receive external impulses; nevertheless, they “know” when it is the time to die and follow the innate program of death. Most probably, processes of “memory” are involved [31]. Tuber PCD is naturally irreversible, but its destiny can instead be experimentally reverted. The accomplishment of PCD can be interrupted, allowing the cell to survive and assume new characteristics. The exceptional reversibility of this death program to a life one is of particular interest, as it can allow us to identify much of the fundamental metabolism necessary for the start of a new life program, which is very poorly understood. Thus, it is necessary to know in depth the condition of the dormant cells, as dormancy is not a unique homogeneous stage, but it is differently responsive to the treatment. This program change can be obtained in the lab by treating the storage dormant parenchyma of the medulla of the tuber of *Helianthus t.* by an in vitro technique and PA supply. This system was the object of a series of research studies conducted since 1965 [23] until recently, almost exclusively in our lab (utilizing the cv. OB1); thus, the literature references are necessarily restricted mainly to some of our articles and completed with data from literature cited throughout the text.

The research was focused on PAs, at that time considered promising growth factors exclusively in humans and bacteria, and on their interaction with fundamental components of the cell, like nucleic acids. The initial target of this research was to ascertain if PAs were possible effectors of the program change, acting solely as growth factors, but later, as our investigations progressed, we delved into their additional roles [16,21,32,33,34,35,36,37].

In the realm of plant biology, the initial recognition of PA as key players in growth processes emerged through the utilization of the *Helianthus t.* model in in vitro culture. Due to its importance for PA studies, it is described in detail in this review. This plant model studied in the lab of Roger J. Gautheret by using auxins was further utilized by Nello Bagni of the Bologna University. The crux of this investigation lies in the substitution of the well-known plant hormone auxin (IAA) with spermine (Spm). Remarkably, Spm exhibited growth effects similar to IAA, albeit with distinct optimal concentrations [23]. This system is unique, as, being formed by a dormant tissue with an exceedingly low PA content, it allowed researchers to also study the effects of other exogenous PAs which, as reported in the literature, acted as a sole nitrogen source for the survival of the explant [38]. This experiment is possible also by adding natural extracts found to contain PAs, for example, orange and tomato juice or coconut milk [39,40]. This assay is not possible with non-dormant plant cells. In fact, the PAs’ natural content, relatively high in a non-dormant living organism (ranging between µM and mM), masked and inhibited the growth effects of the supplied PAs. After this first report on the growth-promoting properties of PAs, a cascade of papers on growth and development phenomena were published in other plants and reviewed by some authors [41,42,43,44,45].

The research from this low-level PA-containing system that Bagni developed for more than forty years [46] was further conducted by several methodologies and produced a large array of results including with applicative targets. As an illustrative example of its applicability, this system was employed to investigate the effects of the growth inhibitor coumarin on cell proliferation [47]. Subsequently, it was also utilized to explore the impact of agmatine, as well as platinum- or palladium-derived drugs, in human cancer cell lines as antiproliferative and cytotoxic agents [48].

The unusual possibility to revert PCD and the identification of some of the mechanisms involved in this change, to restore the characteristics of life, are discussed in the present review to provide some glimpses of knowledge on the role of PAs as essential factors involved in the articulated and not easy subject of PCD and differentiation.

## 3. Main Data from the Research on the Entire Life Cycle of *Helianthus tuberosus*

The tuber was the object of research during its entire life span either under the natural death program (A—Death Program) describing the tuber formation, dormancy and its changes under PCD and also when the tuber was induced to resume a new life program (B—Life program); this part describes either the results obtained during the synchronous amitotic cell cycle of the short-term in vitro culture or the results obtained by treatment with PAs, comparatively with growth substances and inhibitors, to evaluate the growth of long-term cultures.

### 3.1. The Tuber Death Program

The properties of the tuber have been examined in the phases of the three main stages: formation, dormancy and sprouting. At the end of dormancy, the first events of PCD begin. The most significant biochemical features are summarized in Table 1.

The reported data were taken from the cited literature, as follows:Fresh Weight: from all the following references, as this parameter has always been measured.rRNA: [49].Total Proteins, 38 kDa protein, 177 kDa protein: [50].Spm, Spd, Put: [25,48,49,50,51]. These data have been confirmed by [20].ODC ADC, SAMDC: [46,52,53,54].DAO: [55,56,57,58,59]. Confirmed by: [20].Glutamine, Arginine: [60].Bound Spm, Bound Spd, Bound Put: [25,26,27]. The data on polyamine conjugation to proteins in plants were further confirmed by several research studies on the catalyzing enzyme transglutaminase [61,62,63,64].Spm Bound to rRNA, Spd Bound to rRNA, Put Bound to rRNA: [65,66,67].

### 3.2. Tuber Formation

The vegetative shoot growth occurs during the spring–summer months (Figure 2). In the temperate climate of the northern hemisphere, after the plant flourishing, in September–October, the tubers begin to grow from subterranean branches.

Regarding the detailed description of the tuber formation process, the following is reported: the cells of tuber medullary homogeneous parenchyma (Figure 1D) rapidly divide transiently only in a limited central area which then greatly expands so that the tuber total fresh weight increases by more than 300% in two months (Table 1, (*1*)), then remaining constant [48,49]. During this short division phase, the Put biosynthetic pathway occurs via ODC and is only transiently active (not detectable in Table 1), thereafter remaining almost inactive (Table 1, (*9*)) [52].

These cells accumulate different compounds: glycosidic substances, as inulin and fructose, but notably few starches. The expression of numerous genes involved in fatty and unsaturated fatty acids metabolism and plant hormone signal transduction are closely associated with the development of tubers; ethylene-response and heat shock gene families are predominant [68].

The occurrence of a synthetic metabolism is suggested by the increment of rRNA (Table 1, (*2*)) and, similarly, of total proteins (Table 1 (*3*)) as well as that of the most abundant protein of 38 kDa (Table 1, (*4*)). Also, the two free PAs, Spd and Spm, and the activity of their biosynthetic enzymes, namely ADC (arginine-decarboxylase) and SAMDC (S-adenosyl decarboxylase), abruptly increase about 50-fold (Table 1 (*10*,*11*)) [53]. These data show that the two Put biosynthetic pathways, by ODC and ADC, are differently required, according to the cell biological conditions, as ODC is active only during cell division, whereas ADC is active only during cell expansion. Put is not accumulated, even though ADC is very active (Table 1 (*10*)), but it is probably totally metabolized by SAMDC for the Spd and Spm synthesis (Table 1 (*6*,*7*)); this occurs only until the end of tuber formation, when the latter two PAs rapidly decrease and Put content slightly increases (Table 1 (*8*)).

It is possible that new free PAs are utilized no more as free, their role being accomplished in bound form. This is supported by the increase in some bound form at the following initial dormancy phase [25].

In summary, the take-home message at the end of this part is that PAs are deeply involved in tuber formation, in agreement with the general knowledge that PAs are active in growth, but also adding the suggestion of the importance of their connection with proteins and RNA [23,50,65,66], as ascertained in the next stage of tuber dormancy.

### 3.3. Tuber Dormancy

After October/November, tubers cease to grow, as shown by the increase and then constant amount of fresh weight (Table 1 (*1*)), and enter dormancy [51]. By morphological observations, it appears that during deep dormancy, cells of this homogeneous storage parenchyma are inactive in cell division (Figure 1D). Dormant parenchyma cells are characterized by a thin layer of cytoplasm adherent to the cell wall, which includes heavily stainable nuclei also adherent to the cell wall, pressed by a large vacuole (Figure 1D) [68]. Nuclei have 102 chromosomes with a hexaploidy karyotype and small nucleoli. The distribution profile of DNA stainability shows different frequency classes [69,70,71]. The cytoplasm shows small mitochondria [72], suggesting that respiration is also low, and non-photosynthetic proplastids, which contain heavily stained membranous tubular complexes (*TC*) in different shapes [73], instead of the organized grana thylakoids of chloroplasts (Figure 1E).

Regarding the detailed description of the tuber dormancy features, the following is reported: in the winter middle dormancy, the weight is stationary and the large tuber parenchyma cells (arrested in the G_0_ phase of the cell cycle) have a steady percentage of water. The metabolism is slow, despite the high rRNA level (Table 1 (*2*)), as shown by the failure of the amino acid incorporation of the ribosomal fraction if exogenous PAs (Put and Spd) are not added to the assay at an appropriate Mg^2+^ concentration [65,66]. This result is supported by the fact that ribosomes are present mainly as monosomes [66].

The tuber, during the progression of dormancy, seems to accumulate several of the compounds studied. The high level reached by rRNA (Table 1 (*2)*) agrees with the protein increase (Table 1 (*3*)) [65]. In the middle dormancy phase, total proteins and also a 38 kDa one, present in a high amount, as well as a transient main 177 kDa one, all reached the maximum (Table 1 (*3*,*4*,*5*)).

The amount of total free PAs during dormancy increased (Table 1 (*6*,*7*,*8*)); their precursors, arginine and glutamine, were also accumulated, reaching a maximum level (Table 1 (*12,13*)). The bound Put and Spm were mainly present in lower percentages than Spd (Table 1 (*14,15,16*)). They initially increased, then remained stationary for a long period.

By analyzing PAs bound to nucleic acids, namely rRNA (Table 1 (*17,18,19*)) and tRNA, their concentration results are very low during the initial stage of dormancy, whereas they increase 2- or 3-fold in the middle period of dormancy, especially of Spm decreasing thereafter [23,63].

During the final stage of dormancy, free PAs increase or are stationary (Table 1 (*6,7,8*)), whereas the percentage of bound forms is stationary or decreases (Table 1 (*14,15,16*)). In this period, the in vitro protein translation is active without any additional supply of PAs. This suggests that the machinery of protein synthesis is complete in appropriate composition.

Taken together, these data, show the trends of some parameters during dormancy progression. These data provide evidence that dormancy is not a real period of physiological dormancy, in which the metabolism should be almost stopped, as the morphology of the identical cells leads us to suppose. Instead, some metabolism, especially regarding proteins, rRNA, and bound PAs, is active, possibly to prepare the cells for the next expected stage, namely PCD.

### 3.4. Tuber Sprouting and PCD Natural Destiny. (A) Death Program

Whereas in the final phase of dormancy, in which many parameters increased, in the following period the general tendency is to decrease except for PAs (Table 1). Morphological observation shows that, randomly, some of the tuber cells, either when treated with Tunel staining or by immune-staining analysis of the TGase reaction products, show dark-stained pyknotic nuclei suspended in the center of the cell, suggesting that these cells undergo cell death, whose cell wall is definitely marked (Figure 1F,H,I); in addition, some reticulate tracheids are differentiating, showing pyknotic nuclei stained with Tunel stain (Figure 1G). The tracheids should differentiate to favor the transfer of storage substances, and perhaps also of PAs, from tuber to buds, which began to develop forming a sprout (Figure 3A). TGase is known to be involved in the mechanism of PCD [10,15,19,74,75,76,77]. Both these TGase- and PCD-stained cells match their random distribution and appear interspersed among cells with normal nuclei (Figure 1F,H). This could be the initial phase of the next more generalized PCD, involving the entire parenchyma’s terminal differentiation and innate PCD. During this period of the natural release from dormancy (Table 1 and Figure 3A,B), the generalized death of tuber cells proceeds, without showing too much water loss, as shown by the stationary fresh weight (Table 1 (*1*)).

Regarding the details of the tuber sprouting features, the following is reported: their cells are programmed to become depleted by storage substances and rRNA (Table 1 (*2*)) [65]. The total protein content rapidly decreases (Table 1 (*3*)). Already at the end of dormancy, free PAs start to increase (Table 1 (*6,7,8*)) and Spd and Spm continue to increase to a very high amount (Table 1 (*6,7*)), concomitantly with the drop of their possible precursors, glutamine and arginine (Table 1 (*12,13*)). Then, in the final phase of tuber sprouting, namely during PCD after a marked total decrease, the percentage of bound PAs increases again (Table 1 (*14,15,16*)), reaching a level similar (50%) with that of free PAs [26]. By analyzing the bound PAs, it was observed that the PA partners consist of different molecules, as suggested by the treatment with DNase, RNase and proteases of the insoluble pellets of tuber parenchyma slices incubated with ^14^C-Put. In fact, most of the radioactivity is released from proteins (35–55%) and to a lesser extent from NA. RNase mainly releases Spm, whereas DNase releases mainly Spd with respect to the other two PAs [22,23]. Spm and Put were the main PAs bound to rRNA or to tRNA. In February, PAs bound to NA increased considerably, especially Spm bound to rRNA which in one month increased ten-fold; this probably allows amino acid incorporation without the addition of exogenous PAs by the ribosomal fraction extracted in this period, contrary to what was observed during deep dormancy. Also, PAs linked to rRNA and tRNA (not reported) are relatively high, suggesting that protein synthesis might be active.

In addition, free and non-tightly-bound PAs have been found located in the cell walls. Also, an indication of the possible involvement of TGase in modifying cell wall proteins by tightly bound PAs was observed in *Helianthus t.* cycling cells (see below). PAs bound to very high molecular mass substrates were released by cellulase, pectinase and protease, showing that PAs are a component of the cell wall structure [75].

At the end of dormancy, active metabolism begins to take place (Table 1) and these relevant metabolic events are possibly related to the requirements of storage substances by the sprouting of apical buds. During the next sprouting, buds begin to develop, giving rise to roots and sprouts (Figure 3A,B). Their growth needs the transfer from tubers of nutrients, proteins, growth factors, probably PAs and other substances. The sprout apices, notably their meristems, necessitate a considerable amount of PAs for cell division, which does not occur if the PAs concentration is below 10 µM, and for the further development into a mature plant [76]. In the latter, the PA concentration was found in the order of mM, even though most of the PAs are probably stored and not active in the metabolism.

A transfer of PAs has been observed either at long distance in an adult plant through the xylem and phloem [78,79] or intracellularly among different cell compartments [79]. As an example of the latter, Spd is uptaken by *Helianthus t.* isolated mitochondria, depending on the membrane potential and respiration [80]. This uptake occurs even if PAs and their biosynthetic enzymes are already present in mitochondria [71]. PAs are thus accumulated, either by biosynthesis or transport.

In sprout apices, a TGase-like activity was found for the first time in plants [60,61], further confirmed as due to different TGases, some of which have molecular weights and amino acidic compositions homologous to those of animals [44,81,82]. Also, in the parenchyma cells of the tuber of *Helianthus t*., TGase activity has been identified [83]. The tuber total proteins also should be transferred to buds, as their decrease could suggest (Table 1 (*3*)). However, at least in part, they should be directly assembled or disassembled and resynthesized in a 150 kDa one whose amount increased in the dying tuber [50]. This protein incorporates Spd, eventually conjugated by TGases, as well as the other two PAs, all found bound to proteins (Table 1 (*14,15,16*)). For details on each single PA, see the literature [24,25,26].

The binding of PAs to proteins could be non-covalent or covalent. The first is not stable, thus making it difficult to study, even though PAs non-covalently bound to solubilized plasma membrane proteins have been detected in much higher concentrations than the covalent ones [52,53,82]. The main membrane proteins bound to Spd are a 60 kDa and a 18 kDa one, possibly intermediate in membrane fusion and the transmission of signals [83,84].

Once the complex transfer of the useful molecules from the tuber to sprouts is accomplished, the tuber dies, having completed its duty to allow the survival of the plant during the adverse seasonal conditions. Buds then germinate, forming a new plant that is photosynthetically active, thus ensuring the completion of the life cycle (Figure 3A,B).

Summarizing, the natural life of the tuber, even though its cells have genetic and morphological homogeneity, presents a very different metabolism in its three phases: formation, dormancy and sprouting, the latter being a long period constituted by senescence and PCD. The current data open a window on the need for the synthesis of new proteins and a high percentage of bound PAs for PCD to be complete.

### 3.5. Break of Tuber Dormancy and In Vitro Cultures. Destiny Change: (B) Life Program

The tuber PCD can be avoided and reverted by a laboratory intervention changing this death program into a life one, as described below. This consists of explanting the dormant parenchyma and cultivating it in vitro; wound stress, together with the supply of growth factors, will stimulate growth under in vitro culture conditions (Figure 3A,C) [23,85,86]. These cells dedifferentiate and several of their metabolites are degraded. By undertaking new differentiation, the cell behavior can be studied without the interference of the metabolites typical of PCD. These cells must synthesize ex novo the entire molecular kit necessary for the new, different life programs.

Shortly, tuber cells change their death program to a life one, a sort of resurrection. Due to their characteristics, their development was studied in two periods: the very first and detailed one, namely within one day (short-term experiments), and the further one of growth and differentiation, during the following month (long-term experiments). The explants were stimulated with different growth substances and compared to the untreated ones. In particular, the effects of IAA, 2,4-D and other substances metabolically related to PAs were compared to those induced by PAs in order to better clarify PAs’ features.

#### 3.5.1. Short-Term Experiments—Cultures Untreated or Treated with PAs or Growth Factors. The Synchronous Amitotic Cell Cycle

These data derive mainly from results reported by [52,56,58]. The explants taken from the homogeneous tuber internal parenchyma during deep dormancy (Figure 3D–H and Figure 4A) can be cultivated either in a liquid or solid medium [87,88,89,90,91]. For short-term experiments, thin cylindrical slices (0.9 cm in diameter, 0.1 cm-high) are used in order to expose a large surface in contact with the liquid growth medium under agitation for 30 h (Figure 3C, left). These cells undergo multiple abiotic stresses, such as wounding, new nutritional, eventually hormonal, molecular signals and chemical–physical status, mainly the temperature and oxygen increase, the mechanical conditions and the space-position, as well as the water and light availability.

##### Untreated Cultures

The explants not treated with growth substances, but only with the growth medium, do not show any visible modification, and their parenchyma cells do not divide. Simply put, their outermost layer begins to impermeabilize by suberification of their cell walls, which will be easily visible later on [72].

##### Treated Cultures

To study their effects on the explants, Spm and Spd were added at their optimal concentrations into the liquid medium [20,48,50]. An inner cylindrical layer of cells, located at a distance of about ten cell layers from the surface, synchronously enters cell division during 30 h of activation. These cells are in the optimal microenvironmental conditions for cell division. Being at the G_0_ stage, they change their fate (the natural destiny would have been to die), de-differentiate, acquire meristematic characteristics and then manifest their pluripotency. This layer of cells simultaneously reacts to abiotic and biotic stress by entering a synchronous cell cycle divided into the following phases: G_1_, S, G_2_ (virtual) and D (division “D”, a term used instead of the term “M”, mitotic). S and D are partially superimposed. The long G_1_ phase can be divided into two parts (I and II), based on metabolic events (Figure 5B). ^3^H thymidine autoradiography and DAPI staining allow us to identify that, in the G_1_ phase of the cycle, DNA synthesis occurs either in mitochondria, contemporaneous with respiration enhance, or in plastids, concomitant with light exposure, whereas nuclear DNA synthesis starts independently later, at around 12 h [66,68]. This G_1_ phase is characteristic because it not only involves the preparation for DNA synthesis, but also the elimination of aliquots of proteins and NA, probably typical of dormancy. The innermost cells of the explant do not divide [66,86].

The length of the cell cycle depends on the explant shape and volume, as well as on the deep dormancy phase of the tuber and other factors, such as, for example, the homogeneity of the tissue.

(A)Morphological aspects of the synchronous cell cycle of treated cultures.

Among the first morphological events observed after the explantation, there is, in a peculiar ring of cells, a shift of the flat nuclei adherent to the cell wall towards the center of the cell, assuming a round shape; their small, numerous nucleoli [90] seem to coalesce in a unique nucleolus inside a nucleolar vacuole or cavity (Figure 4B,C). Some nuclei embed the nucleolar organizing region of the nucleolar chromosome (Figure 4D, the detail “L”) which enlarges until 15 h and then disperses throughout the nucleolus. The nucleolar volume increases until 19 h, in connection with a possible amplification of rDNA and the synthesis of ribosomes. Other nuclei extrude nucleoli or nucleolar bodies, among which, possibly the light-sensitive ones, developed when the explants suddenly were exposed to light (Figure 4D,E) [69]). These morphological complex events reflect the RNA metabolic changes described below.

Neither signs of nuclear membrane solubilization nor condensation of chromosomes nor of mitotic spindles are visible, also by treating the cells with colchicine and vinblastine which also inhibit PA synthesis. Instead, the nuclear membrane insinuates in the nucleoplasm, thus dividing the nucleus and its nucleolus into two new nuclei that contain similar DNA amounts (Figure 4F–H) [72]. This type of division is called “amitosis”. Soon in these binucleate cells, a new thin cell wall, oriented mainly parallel to the explant surface, arises from the mother cell wall. It divides the cell completely into two parts, internal and external, along a ray, each with a uni-nucleolar small nucleus (Figure 4I,J). The orientation of the new cell wall causes the formation of a thickening cylinder of new dividing cells. The frequency classes of the distribution profile of DNA stainability which were numerous during dormancy, were reduced to a single sharp peak; thus, the DNA content is made homogeneous by amitosis division. The innermost medullary cells of the explant do not divide [66,71,86,88].

Thus, growth substances are necessary factors for growth but the microenvironment, to which cells are exposed, conditions their development. Relevant is that this type of division is very peculiar, consisting in a first amitotic cell cycle also reported for tobacco cells [94]; in these cells, any characteristic of the mitotic division fails. It has also been observed in other in vitro growing tissues and interpreted as “first aid” division to rapidly repair a wound [71].

(B) Physiological aspects of the synchronous cell cycle of treated cultures

It being too difficult to discuss in detail all the physiological representative data of the entire cell cycle, the principal of these have been summarized in a temporal schema of the synchronous cell cycle in Figure 5, which is composed of three parts: A, B and C.

A, Dormant Tuber.

The square insert (Figure 5, Part A), summarizes the main quantitative characteristics of the tuber during deep dormancy, already partially described. In the square insert, the main events are summarized: the metabolism of PAs, as well as their content, and that of NA, which is low during the G_0_ phase, whereas the PAs precursors are at a high level.

B, Synchronous Cell Cycle.

Transition from G_0_ to G_1_ Phase.

By activation of the tuber explant, the G_0_ cells enter the G_1_ phase and their synchronous amitotic cell cycle begins (Figure 5, Part B). A peculiar aspect is the immediate degradation of many compounds after activation. Some of the most abrupt and precocious detectable events regard PAs. The degradation of a conspicuous amount of arginine, asparagine and glutamine, the main amino acids stored, was observed within a few minutes after activation; they are immediately followed by the synthesis of PAs and their metabolites and of the TGase activity [82,95,96]. PAs are thus conjugated by TGase (whose synthesis continuously increases) to proteins, which also undergo an immediate decrease, followed by a new increase. NA also immediately reacts to activation: RNA, within 30 min, and DNA, within 1 h, show a decrease [97,98], whereas, contemporaneously, mtDNA and ptDNA, as well as the total RNA, are synthesized, and the latter continues until the mid-I part of the G_1_ phase. Within 3 h, PAs bind to tRNA and especially Put to rRNA, tRNA and a poly-A-containing fraction. Nuclear behavior has been analyzed: during dormancy, nuclei, which have several nucleoli, are flat and adherent to the plasmatic membrane and cell wall. Differently, at the onset of the cell cycle, they move into the center of the cell, become spherical and the nucleoli fuse into a single large nucleolus [72]. PA synthesis then increases, showing a maximum at 12 h, concomitant with the beginning of chromosomal DNA synthesis and the increase in one of the PA degradative enzymes, DAO [58,85,90]. Before 3 h, polysomes are disaggregated and soon reconstituted. This shows that the protein synthesis, which was very low during deep dormancy, starts again, probably after the degradation of part of its previous machinery and products [66]. Several compounds are thus synthesized and degraded during the first half of the G_1_ phase. Along phase II of G_1_, many enzymes involved in PA metabolism increase.

In summary, the G_1_ phase is considerably long (12 h), but it comprises not only the preparation of the DNA synthesis, as occurs in the normal mitosis, but also a period in which the molecules, typical of the dormancy period (A), are no longer useful and indeed are degraded. This represents the critical period of change from the storage and PCD program to prepare the cell for the new life activity.

S Phase.

At the beginning of chromosomal DNA synthesis, namely at 12 h, the PAs (Put and Spd) content, which is at the maximum level, rapidly follows a decrease, possibly caused by a maximum of DAO activity, even though most of their biosynthetic enzymes are still active (ODC, ADC, SAMDC, ARGase) [57]. Alternatively, free PAs can also be “sequestered” by binding to some molecules, for example, DNA or proteins; in fact, the binding of PAs constantly increases. The amount of the latter moderately increases up to 20 h, but decreases up to phase D. TGase synthesis increases, as confirmed by the continuous synthesis of 58 and 90 kDa immunodetected proteins.

D phase.

When the cells divide, PAs, after a maximum, especially of Put, begin to decrease, even though DAO drops. This decrease could perhaps also be correlated with the fact that part of the PAs is no longer measurable as free, as a consequence of linkage to other molecules, like NA; proteins, for example, cytoskeletal and cell wall ones; pectic substances; or others. The observed formation of several new cell walls (Figure 4I,J), of which PAs are known to be components, and the fact that TGase conjugation also remains very active (Figure 5, Part B) could support the hypothesis that PAs are sequestered by the growing cell walls [75,76].

Summarizing, the PAs amount and their binding appear variable, strictly depending on the phases of the cell cycle, suggesting that they have a functional role in its important events. In fact, PAs form linkages with fundamental molecules suggesting that they have numerous roles in protein synthesis, chromatin organization, transcription and translation, but also in post-translational events. Most of the above data were reported in literature [54,55,63,97,98,99,100,101,102].

Some experiments were performed to confirm these suggested roles. By incubating the cells during the cell cycle with labeled PAs, it was discovered that PAs bound to proteins concomitant with their synthesis, thus suggesting that they could be related to each other [76]. In particular, very high molecular mass proteins quantitatively increase during the cell cycle and incorporate either labelled methionine (by 10 times) or Spd, when assayed in the G_1_ or G_2_/D phases. The binding of PAs to proteins is an event that frequently involves the post-translational formation of covalent linkages mediated by TGase. In particular, two peculiar bands of 17.5 and 19 kDa proteins incorporate Spd and decrease between 12 and 15 h in coincidence with the beginning of nuclear DNA synthesis [50]. These bands are interpreted to be the elongation factor eIF-5A modified by hypusine, an amino acid derived from Spd [84]. Alternatively, they are identified as a type of histone.

PAs are also related to cell wall formation, as they covalently link to its proteins and also to cytoskeletal proteins. Of course, other effects that were not examined are possibly involved, like, for example, that PAs, as precursors of GABA, could also furnish organic acids for the Krebs cycle. In addition, PAs are directly bound to NA (Appendix A). As a confirming example, by adding Spm in vitro to the complex DNA–actinomycin D, the absorbance of the complex changes. This suggests that Spm displaces actinomycin D from its binding to the double helix, an event which inhibits the biological activity of DNA, as it occurs in cancer [102,103]. Moreover, in activated tuber cells, Spd stimulates DNA synthesis, fairly doubling it in respect to the actinomycin D-treated cells, also stimulating the synthesis of RNA [103].

C, Other cell cycles.

In Insert C in Figure 5, the patterns of either PAs or of some of their biosynthetic enzymes (ODC and SAMDC) during the cell cycle of *Helianthus t.* are superimposed to those of the mitotic cycles of Hamster ovary cells (CHO) [92], *Nicotiana tabacum* [93] and *Catharanthus roseus* [94]. They show a bimodal pattern: the two maxima of both enzymes precede the two maxima of free PAs, whose minimum coincides with the middle phase S. This decrease could confirm what was suggested above, namely that perhaps PAs are not free, becoming linked to some partners, like NA or proteins [37,50,66,84].

It is relevant that also in synchronous cell division of other organisms, PAs, as well as some of their biosynthetic enzymes, present a pattern related to the different phases perfectly superimposable to that of *Helianthus t.* [85,86]. This coincidence shows that PAs behavior is not peculiar only to the amitotic synchronous cell cycle of *Helianthus t.*, but could pertain to a general mechanism of division independent of the organism involved and the type of cell division.

Summarizing, the variations in PAs during the cell cycle, frequently in bound form, are strictly connected to those of NA and proteins, possibly favoring their functions.

#### 3.5.2. Long-Term Experiments—Cultures Untreated, Treated with PAs or Growth Factors, Promoters or Inhibitors

For long-term experiments, cylinders of parenchyma (0.3 cm in diameter; 0.3 cm in height) were inserted with their basal part in the agarized solid medium. Different conditions were assayed: the explants were treated only with media (untreated cultures) or with media supplemented with different PAs, various metabolites of PA biosynthesis or inhibitors (treated cultures) for 30 days, under light/dark conditions.

##### Cultures Untreated with Growth Medium, without Growth Substances

(A)Morphological observations

The explants of the untreated cultures (Figure 6A) do not divide; their outermost cells only protect the culture by modifying in a few days the cell walls from the external environment (air or medium). Explants are less soft with respect to treated ones, and the cells have thick walls with intercellular spaces full of optically dense material (Figure 6B). Cohesion between cells is demonstrated in animals to be due to an extracellular TGase; a similar event could occur in these untreated cultures. The existence of an extracellular TGase was detected in growing pollen as an active component of the cell wall [104]. Nuclei showed no division and had a similar DAPI-stained fluorescence.

The untreated explant presents an evident greening (Figure 6C–F). In fact, an important modification in the untreated explant is the transformation of their proplastids into chloroplasts, observed to begin on day two. In this time phase, the tubular complex (Figure 1C (*TC*)), present in the explant at time 0 (where also protein crystals and clusters of phytoferritin particles are visible) (Figure 6C), begins to enlarge accumulating pigments and become more heavily stained, showing a strict association with mitochondria (Figure 6D). The tubular complex soon occupies almost all the volume of the plastid and the number of starch granules increases. In a one-week-old untreated explant, the plastids develop a few parallel thylakoids (Figure 6E), which, after three weeks, form appressed small grana, whereas the tubular complex was no longer present (Figure 6F). This differentiation is similar to that observed in several plants during natural proplastids’ evolution to chloroplasts [105].

(B) Physiological features

Parallel to the morphological observations, a physiological study conducted on TGase activity shows that it increases significantly within a one-week culture. In this period, chlorophyll is synthesized two-fold more rapidly in untreated than in treated samples. In order to evaluate the TGase activity, the incorporation of [^3^H] Put was measured in the cell-free extracts of explants: the untreated sample, grown for one week, incorporated two-fold more (0.510 nmol/explant/h) than the treated explants (0.270); a higher difference is visible if expressed per mg prot^−1^. The Put conjugation was analyzed by SDS-PAGE: the radioactivity is considerably higher in the untreated sample especially on the charge of high molecular mass proteins, but also in some low molecular mass ones; among the latter, one co-migrates with the large RuBisCo subunit, in contrast to those of the treated sample, which do not show any label in this electrophoretic region. These data support the hypothesis that many neo-synthesized proteins, components of developing chloroplast structures, can be substrates of TGase, much more actively in untreated explants [105,106,107,108]. The chloroplast TGase activity is involved in photosynthesis and stimulated by light, as also shown in isolated *Helianthus* chloroplasts [106,107]. Margosiak and collaborators [109] found that the large RuBisCo subunit is a TGase substrate. More recently, it was shown that many other chlorophyll-proteins are TGase substrates involved in photosynthesis and photoprotection [95,106,107,109].

These were the first data showing that TGase is a compartmentalized, light-sensitive enzyme and they greatly enlarge the knowledge of this enzyme, whose features are plant-specific.

##### Cultures Treated with PAs, Growth Factors Promotors or Inhibitors

The treated explants initially divide amitotically as reported above (Figure 4), then develop during 30 days and produce differentiated tissues. But, after some synchronous cell cycles, the treated cells begin to divide mitotically in an asynchronous series of cell divisions (Figure 6G–O). Thus, the two interesting characteristics of synchronicity and amitosis are lost and the identification of some physiological parameters becomes difficult. Thus, they were no longer analyzed day by day, but mainly the morphological events were observed during the growth to see how the neo-synthesized cells were able to differentiate. At the end of the period of growth, the effects of PAs, of their inhibitors and other growth substances were evaluated.

In the explants treated with growth substances [66,67,88], after the first division, within two days, one of the daughter cells again divides into two cells; the other cell shows the nucleus suspended in the middle of the cell preparing for another division (Figure 6G). Later, after five days from explantation, the original large cell had already divided into seven smaller cells; it is possible to observe in some sister cells the small nuclei leaned against each other at the opposite parts of the new cell walls (Figure 6H). These examples show that the large initial parenchyma cells acquire the dimension of small dividing cells. In fact, in the subsequent days, the cells located in the dividing cell layer (Figure 6I), deriving from the initial one observed during the first cell cycle, further divide mitotically. Then, they differentiate into conduction cells and aggregate with other cells, the phloem and xylem, arranging into collateral bundles (Figure 6K). The latter are anatomically organized according to the eustelic stem-type, an anatomical structure typical of eudicotyledons, like in the *Helianthus t.* adult stem (Figure 6L,M). In fact, the explant “remembers” its stem origin. How this occurs is pure speculation.

Other cells differentiate according to their position. The ones that are external with respect to the ring of bundles give rise to the dividing cortical parenchyma and the most superficial ones to a protection suberized layer (Figure 6J) differentiating at the end of the culture (Figure 6M). The other cells, internal to the vascular ring, remain unmodified or sometimes differentiate in unorganized aggregates, formed by long filaments of tracheids without a precise direction, seldom associated with sieve tubes, or form periphloematic or perixylematic nodules (Figure 6N,O). These last formations develop depending on their position in the explant. The availability of growth substances, in fact, is higher in the basal part in contact with the medium, and their concentration is too high to allow a regular development. The concentration of the growth substances along the explant shows a gradient dependent on the time of culture, efficiency of the transport and distance from the source, namely the growth medium. The explant does not develop in a perfect cylinder, becoming larger in the apical part (Figure 7C).

What is described takes place only in the samples treated with growth substances (included PAs). These explants enlarge, become friable, synthesize many proteins and greatly increase their weight: after one week it is already more than doubled, whereas the weight of untreated ones increases only by a few weight percentage points. An example of the difference between the explants at the end of the experiment is shown in Figure 7A,B.

The different behavior of untreated and treated explants shows that general growth and chloroplast differentiation are regulated by different programs. Both explants are alive in the same medium containing the same nutritive substances. It cannot be excluded that growth factors also affect the capacity to uptake nutrients, which condition the energy available.

To evaluate the growth effect of PAs compared to the classical auxins, the four most common PAs, namely Spm, Spd, Put and Cad, were supplied in the medium, showing that the explant growth is similar at the respective optimal concentrations (Figure 7E) [23,39]. Also, the anatomical development is very similar to that of 2,4-D treated explants (Figure 6N,O) as shown by the explant treated (as an example) with Spm, presented in Figure 7D. In Spd, Spm, Put and Cad-treated explants, the differentiation of the vascular bundles is very similar with respect to those of IAA-treated samples (Figure 6M) [89]. In all conditions, in the external cortex, a phellogen formed by several layers of dividing cells is externally limited by a layer of dead tegumental cells (Figure 7D).

Several substances, among which are inhibitors of the PAs biosynthesis, were assayed to verify if PAs are necessary for growth [53,96,110,111]. Two controls, one untreated and another treated with 2,4-D (which has the same effect as IAA containing medium, at optimal concentration), were assayed. The inhibitors of the two main Put biosynthetic pathways (ODC and ADC), canaline and canavanine, have strong inhibitory effects (Figure 7F) so that the explants did not grow and showed some PCD signs. Similarly, DFMO and DFMA, other inhibitors of ODC and ADC, completely block growth, but only when utilized together, thus inhibiting both biosynthetic ways as the interchange between ornithine and arginine, the precursors of ODC and ADC, is active. Methylglyoxal bis-guanylhydrazone (MGBG), which inhibits SAMDC, lowered the Spm and Spd content to 40% within 1 h, but the inhibition disappeared by 6 h; thus, MGBG was probably used in a concentration too low to be permanent. Diaminopropane is a product of Put catabolism, and it is possible that at a high level it inhibits the catabolism of its precursor, Put; thus, Put increases, allowing a considerable growth increment [96].

The complex of these data leads us to conclude that PAs are necessary for the explant growth, whereas other growth factors, like IAA or 2,4-D, are replaceable. As IAA causes the immediate synthesis of PAs (Figure 5B), it can be concluded that the activity of IAA is mediated by PAs. The “historical” picture in Figure 7C, taken already in the sixties, summarizes the conclusion that IAA is not absolutely necessary for the correct growth of the explants, as PAs also produce definitely similar effects (Figure 7C,D). The stimulating growth effect of PAs at the cellular level has also been observed for pollen tube growth [110].

##### Cultures Treated with Drugs and PA-Related Antiproliferative Compounds for Health Applicative Purposes

In human pathology, as well as for economical and ethical purposes, to avoid experiments on living animals, this plant system could have new applications. Specifically, this plant could be used in the evaluation of the cell toxicity and antiproliferative activity of different compounds, both of natural origin or chemically synthesized [48]. Different agmatine analogs in animal tissues are known to act as neurotransmitters or neuromodulators and to display clinical properties such as neuroprotection and tumor suppression. To verify a correlation between their chemical structure and their role as antiproliferative and cytotoxic compounds, as well as their relation to PA metabolism, these agmatine derivatives were assayed in the *Helianthus t.* system. One of these analogs seemed to induce ADC, ODC and OTC activities leading to the production of Put, resulting in the best active compound.

Moreover, considerable effort has also been put into the development of novel metal-based drugs, including cisplatin-like agents, aiming at improved antitumor efficiency [48]. As some polynuclear multifunctional Pt (II) and Pd (II) which chelate with flexible biogenic PAs, as bridging ligands, present severe clinical drawbacks, they have been evaluated by the *Helianthus* system. Cisplatin was the most active both in cell proliferation inhibition and in PA metabolism. Thus, the growth and metabolism of PAs of *Helianthus t.* might be a measure to reveal the characteristics of several compounds not only for basal research but also applicative and medical purposes.

## 4. Concluding Remarks

The results reported by this review are here summarized to synthesize the multiple data reported and point out the main results. It begins with a description of the stages of development of the *Helianthus t*. plant. The first target of this multiple research is the particular attention dedicated to tuber dormancy, its phases of development, dormancy and sprouting until the beginning of PCD according to the natural destiny of the tuber. Its fate is, in fact, to transfer the storage substances to the developing buds, which need them to sprout to form the new plant. PAs are deeply involved in tuber formation, suggesting the importance of their connection with proteins and RNA. The trends of some parameters during dormancy progression evidence that dormancy is not a period in which the metabolism is almost at a standstill, as the morphology of the identical cells leads us to suppose; instead, some metabolism, especially regarding proteins, rRNA and bound PAs, is active to prepare the cells for the next expected stage, namely PCD. The parenchyma cells present a very different metabolism in their three phases: tuber formation, dormancy and sprouting, the latter being a long period constituted by senescence and PCD. A need was observed for the synthesis of new proteins and for a high percentage of bound PAs, both of which are necessary to complete PCD.

A relevant target is also to verify if PCD is reversible, which is an important biological challenge substantially unknown. It is possible to revert the tuber PCD by a laboratory intervention, changing this death program into a life one. Cells in deep dormancy are the best source to perform the experimental induction of the change in their destiny, a sort of resurrection. This was studied in two periods: namely within one day from the beginning of in vitro culture (short-term experiments), and the further one of growth and differentiation, during the following month (long-term experiments). The explants were stimulated with different growth substances to compare to those induced by PAs in order to clarify PAs features that at the beginning of this research were unknown. These experiments are historically relevant as they were in absolute the first demonstration that PAs act as growth factors in plants and also opened the extraordinary development of research in other plants.

The transplanted cells in the G_0_ stage change their fate, de-differentiate immediately, acquiring meristematic characteristics and then manifest their pluripotency. Either from the morphological or physiological aspect, numerous changes take place: their synchronous phases of the amitotic cell cycle allow us to study the reprogramming of the metabolism of fundamental molecules. Abrupt PAs and TGase synthesis and activity occurs and PAs are in part conjugated by TGase to newly synthesized proteins. PAs have several roles in protein synthesis, chromatin organization, transcription and translation, but also in post-translational events. The G_1_ phase represents the critical period of change from the storage and PCD programs to prepare the cell for the new life activity. This system thus has the property to be constituted by PA-low-level cells and thus more susceptible to their supply, and to have a synchronous reaction which allow us to obtain clear metabolic data.

Explants cultivated for long periods divide mitotically, allowing their differentiation to acquire a typical dicotyledonous stem anatomy. The growth effect for the explant of PAs (Spm, Spd, Put and Cad) compared to the classical auxins is similar. PAs are necessary for explant growth, whereas other growth factors, like IAA or 2,4-D, are replaceable. As IAA causes the immediate synthesis of PAs, it can occur that the activity of IAA is mediated by PAs. When the explants are treated with drugs and PA-related antiproliferative compounds, an inhibition of growth has been observed, confirming the observed PA roles. In human pathology, also for economical and ethical purposes, to avoid experiments on living animals, this plant system could have new applications in the evaluation of the cell toxicity and antiproliferative activity of different compounds in tumors.

In the explants not treated with hormones, the development of dormant proplastids in chloroplasts is very interesting. Experiments support the hypothesis that some neosynthesized proteins, components of developing chloroplast structures, can be substrates of TGase. The chloroplast TGase activity is involved in photosynthesis and stimulated by light. These were the first data showing that TGase is a compartmentalized, light-sensitive enzyme, and they greatly enlarge the knowledge of this enzyme, whose features are plant-specific. As reported above, the present review shows that one of the main results obtained is that it is possible to revert the PCD of *Helianthus t.*, which in natural conditions is irreversible, to a new life cell program. Some points related to PCD and PAs to be remarked upon are discussed below.

Several factors, like PAs and plant growth regulators, have been shown to play a critical role in the initiation and regulation of the death process. According to the morpho-physiological state of the target cells, the role of PAs in plant PCD appears to be multifaceted, acting in some instances as delaying molecules, whereas in others they seem to be implicated in accelerating PCD [9]. In both conditions, only the rapidity of the process is regulated, but it is mandatory in any case. In natural PCD, this event is only the terminal part of an entire complex program of differentiation occurring in dying plant organs.

The timing of the death can be regulated by PAs depending on the stage of maturation and differentiation of the organ involved, but not reverted to its initial condition. On the contrary, the present paper shows that the tuber, even when completely differentiated, has the capacity to perform the interruption of its natural PCD and to revert toward different new life programs by PAs. This depends on the receptivity of the organ, which in turn is related to the stage of tuber dormancy and to the relatively simple differentiation of this storage parenchyma. By contrast, PCD inevitably occurs in other types of structures that are much more differentiated. For the *Helianthus t.* tuber system, it is important that the cells are homogeneous. All cells are in a stationary G_0_ phase, are fairly inactive, are all identical, have the same origin, have a scarce PA content and all are in the same environment.

The present data also evidence some aspects of PAs to be remarked upon. A unifying concept to interpret the PA physiological role is still missing, even though a general idea is that PAs are essential juvenile growth substances in all living organisms; however, difficult-to-interpret data are sometimes encountered. This is due to the fact that PAs are present in all cell compartments with specialized functions. They move inside and outside the organelles, rapidly binding to and releasing from other molecules, regulated by local pH, ions, oxidative reactions and a series of other microenvironmental conditions. Moreover, the events of a cell must be coordinated to those of the other neighboring cells, perhaps having different roles. The conditions might continuously change, and the experimental indirect methods used frequently allow only a static picture which should instead be dynamically interpreted.

As mentioned above, PAs can be active not only in free forms, but also in bound ones. Gene overexpression or knockout have helped to recognize and confirm several roles of PAs. As an example, it was possible by using a KO mutant of the TGase gene to verify that some activities attributed to free PAs are instead due to their conjugated forms [64]. In the case of bound PAs, the length of the PA skeleton, which brings the terminal amino groups bound to different other molecules, might explain why the three PAs have different effects. The length of the PA bridge could affect the structure of the partner molecule and, thus, its functionality. In fact, frequently the activity of particular PAs is specific, even if sometimes a PA can vicariate the other ones. This could also be due to the rapid interconversion of one PA into another, sometimes making data interpretation difficult.

TGase activity is generally high in differentiated tissues and low in rapidly growing ones. This also occurs in growing explants of *Helianthus t.*, as shown by these results. In cancer pathology, the animal TGase 2 activity level has become an important parameter related to all the processes of tumorigenesis [18,112].

As PAs directly interact with the most significant molecules of the cell, it can be hypothesized that they do not need other secondary messengers that translate their message to the cell. On the contrary, the classical hormones possibly require PAs to be synthesized as intermediates to exert their effects, as observed for IAA induction of PAs immediate synthesis in *Helianthus t.* in vitro cultures [87].

It can be concluded that PAs are not simply growth substances, like the classical ones specific to plants. In fact, they differ for several characteristics: (1) They are basic and linear molecules; (2) They directly interact with several fundamental cell molecules; (3) They are active in higher amounts with respect to other growth substances; (4) They are synthesized in all cells and their compartments; (5) PAs do not have sites of synthesis different from sites of action (source and sink). Thus, PAs, simple, small aliphatic molecules relatively similar to each other, cannot be considered equivalent to growth factors.

PA effects are sometimes defined as a “mystery” as suggested by some authors [113]. Rather, it should be concluded that PAs form a “multiverse”, namely various coexisting universes, “based on metabolism, ionic interactions, covalent and non-covalent binding, interaction with hormonal pathways and signaling molecules” [114]. It depends, site by site, on the locale and type of PA amount whose effect is strictly related to the particular conditions of the tissue involved. Continuous research on PAs leads to the conclusion that they and their homeostasis play a wide role in plant processes, helping the latter in differentiation, inducing totipotency and increasing cell division. They also play a role in molecular signaling as well as in protecting cells from stress [11,14,115]. Moreover, PAs have also contributed to the in vitro techniques used for culturing plants for commercial or medical purposes. Thus, PAs can be considered a boon to the plant tissue culture field.

These experimental data, and many future data, are the necessary basis to obtain, probably only by artificial computational intelligence, a complete panorama, moment by moment, on PA roles in the cell universe.

The incredible development of the research on PAs accomplished by hundreds of scientists, too many to be mentioned, was unpredictable when, in the sixties, PAs were examined for the first time. They were studied to assess if they could simply be plant growth factors but, rather, further evidence has shown that PAs have many other properties in addition to those hypothesized at the onset of this research.

## Figures and Tables

**Figure 1 ijms-25-05386-f001:**
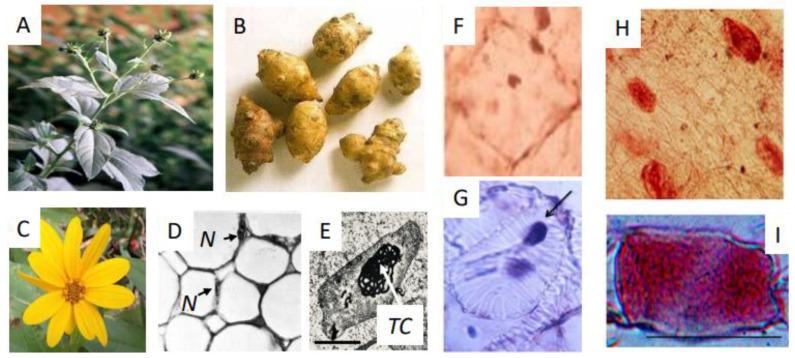
The *Helianthus t.* plant. (**A**) The stem with floral buds; (**B**) The tubers; (**C**) The flower; (**D**) Sections of the dormant parenchyma of the tuber. Light micrograph, semithin sections, N = nuclei [24]. (**E**) Plastids of the dormant tuber containing the tubular complex (*TC*) (TEM × 20,000) [24]. (**F**–**I**) Tuber parenchyma cells in PCD phase. (**F**) Cell with pyknotic nucleus stained by immune cytochemical recognition of TGase-conjugated dansyl cadaverine, marker of TGase [5]. (**G**) Tracheids during their differentiation with pyknotic nuclei evidenced by Tunel stain. (**H**) Panorama of the parenchyma cells incubated with dansyl-cadaverine, a substrate of TGase, and then immuno-cytochemically stained by an anti-dansyl antibody. (**I**) Single cell immune-recognized as in (**H**) [17]. Bar originally 50 μm [5].

**Figure 2 ijms-25-05386-f002:**
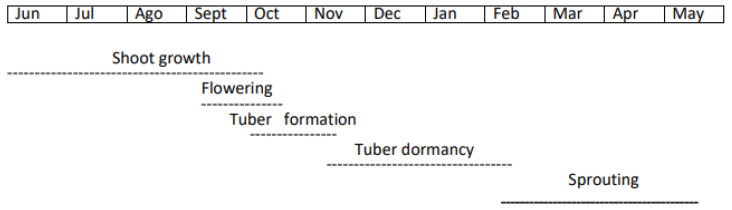
Correspondence between months of the year and stages of development of *Helianthus t.* shoot, flower, tuber and sprout.

**Figure 3 ijms-25-05386-f003:**
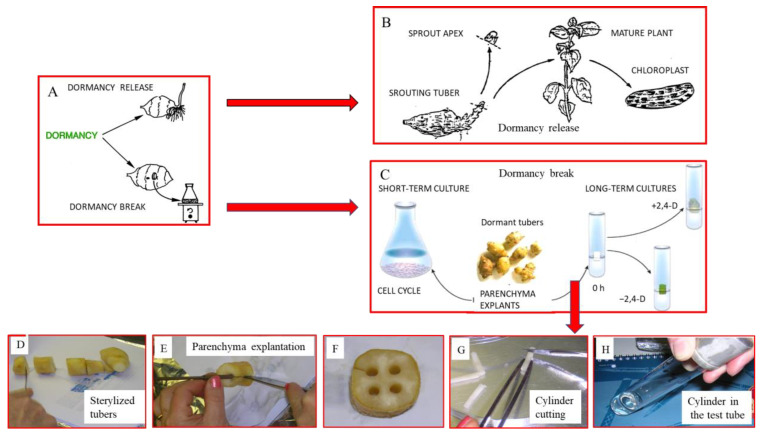
The tuber dormancy and its release or break. (**A**) In February, the dormant tuber in nature releases dormancy and develops an apical bud and roots. Alternatively, in deep dormancy, this stage of dormancy can be interrupted by explanting discs of parenchyma and cultivating them in vitro. (**B**) Dormancy is naturally released. The tuber is naturally developing, producing the growth of the sprout, whose apex has been cut and examined for the first detection of TGase. Normally, the sprout develops and produces the mature plant, whose leaves have been extracted to isolate chloroplasts. (**C**) Dormancy break. Explant discs (0.9 × 0.1 cm) of the dormant tubers can be cultivated in liquid medium for short-term experiments to study the first synchronous cell cycle. Alternatively, parenchyma cylinders (0.3 × 0.3 cm) can be cultured on solid medium for long-term cultures for experiments with or without growth substances, as for example 2,4-D, or other substances. (**D**–**H**) Pictures in sequence of the explantation by a cork borer of parenchyma cylinders in the laboratory.

**Figure 4 ijms-25-05386-f004:**
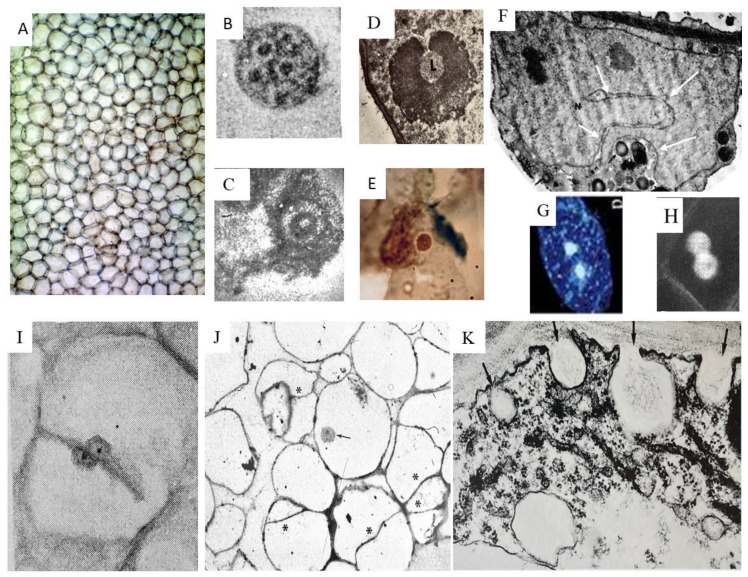
The *Helianthus t*. first cell cycle. Morphological aspects of 2,4-D or Spm-treated explants and exposure to light. (**A**) Medullary dormant parenchyma (light micrograph, ×30) [88]. (**B**) Nucleus with several nucleoli during the early G_1_ phase (light micrograph, ×1000) [88]. (**C**) Coalescence of nucleoli in the late G_1_ phase inside the nucleolar vacuole (light micrograph, ×1000) [88]. (**D**) Nucleolus increases and englobes the light region (L) (3 h of the G_1_ phase) (TEM, ×14,000) [68]. (**E**) Nucleolar extrusion at 18 h (light micrograph ×1000) [88]. (**F**) Nuclear division with deep invaginations of the membrane of the nuclear envelope (arrows) with two nucleoli (TEM ×17,000) [72]. (**G**,**H**) Binucleate cells. DAPI-stained (G ×100), note the fluorescence of chloroplasts and mitochondria, (H ×300) [72]. (**I**) A cell dividing by a cell wall arising from the mother wall with two mononucleolar nuclei adherent to the new wall (light micrograph, ×400) [88]. (**J**) Panorama of parenchyma cells dividing with internal cell walls (thin section, ×75) [72]; * indicate new cell wall; arrow indicates a nucleus. (**K**) Cell cytoplasm close to plasmalemma, with Golgi complex, endoplasmic reticulum and cell wall showing invaginations containing fibrillar material (TEM ×32,000) [68].

**Figure 5 ijms-25-05386-f005:**
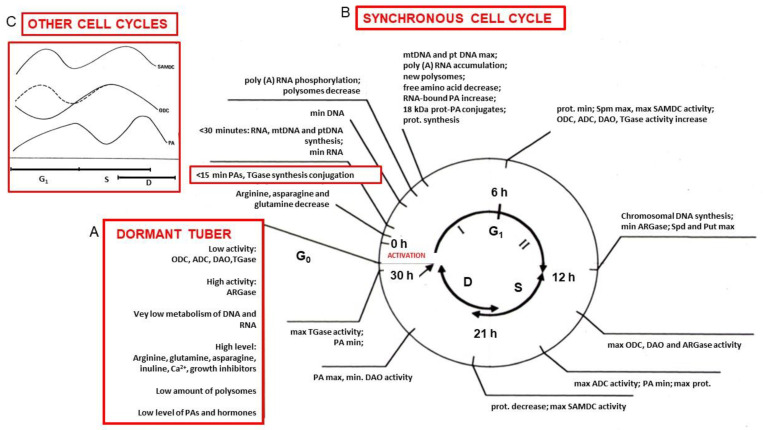
The *Helianthus t.* schematic physiological events of the dormant tuber and explant playing out the first synchronous cell cycle. (**A**) *Helianthus t.* dormant tuber showing the content of various substances and enzyme activities important for the stage of dormancy in G_0_ phase. (**B**) Diagram of the first synchronous cell cycle (0–30 h) of growth-substances-activated explants showing the variations in the main substances and enzyme activities during the various phases (G_1_, I and II parts; S (DNA synthesis) partially superimposed to D (division phase)). Min: minimum; max: maximum; act: activity. (**C**) Superimposed pattern of ODC, SAMDC, PAs detected during the same phases of the cell cycles of other organisms, in detail: Hamster ovary cells (CHO) [92], *Nicotiana tabacum* [93] and *Catharanthus roseus* [94] in addition to *Helianthus t.* tuber [86]. Dotted line in G_1_ phase represents the ODC activity of the cell cycle of some animal cells. Many data are taken from the following literature [14,23,25,31,33,34,36,40,41,42,48,50,54,57,58,66,67].

**Figure 6 ijms-25-05386-f006:**
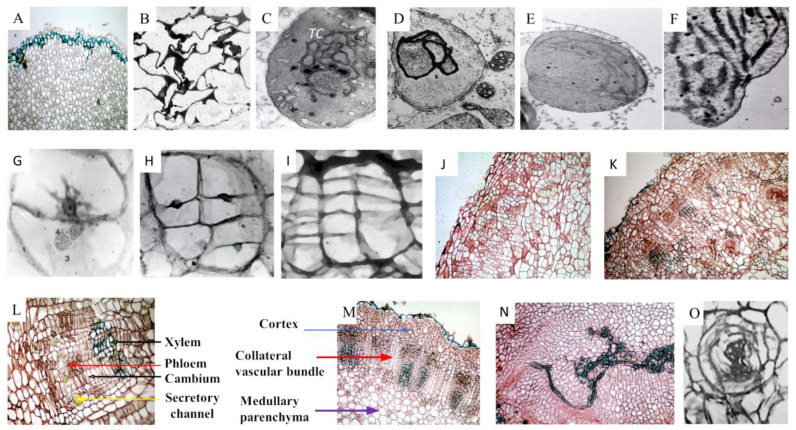
The *Helianthus t.* untreated explants (**A**–**F**) or treated with 2,4-D, IAA, Spd and Spm (**G**–**O**) during long-term growth and exposure to light. (**A**) Untreated parenchyma. Panorama of the explant with the peripheral layers modified after 3 weeks of culture (light microscopy, ×30). (**B**) Untreated parenchyma. The basal part of the parenchyma immersed in the culture medium with intermolecular amorphous material after 3 weeks of culture (thin sections, ×75) [72]. (**C**–**F**) Untreated parenchyma. Development of non-photosynthetic plastids with tubular complex (*TC*) and phytoferritin into chloroplasts. Proplastid (**C**) evolves (**D**) (TEM ×21,000), their membrane aggregates into a single or few membranes in one week (**E**) (TEM ×25,000) and finally into membrane aggregates forming pseudo-grana thylakoids (**F**) (TEM ×36,000) in three weeks (ultrathin sections) [72]. (**G**) Cell of a parenchyma treated with growth substances, divided by an internal cell wall (2) into two cells. One shows a single nucleus with a single nucleolus (above) suspended by cytoplasmic bridges and the other with two nuclei (3 and 4) with multiple nucleoli, still divided after two days of culture (light microscopy ×550) [88]. (**H**) Cell of a treated parenchyma divides into 7 small cells inside the mother cell after five days of culture (light microscopy ×400) [88]. (**I**) Series of small cells derived by internal cell walls’ formation after multiple divisions of a cell of a treated explant after 10 days of culture (light microscopy ×400) [88]. (**J**) External part of the treated explant after 15 days of culture. (light micrographs ×150) [88]. (**K**) Treated explant after 35 days of culture showing an interrupted ring of vascular bundles and, internally, aggregates of tracheids and sieve tubes [88]. (**L**,**M**) Details of the above vascular collateral bundles and their position [88]. (**N**). Examples of aggregates of vascular filaments unorganized, somewhat delimited by sieve tubes, formed in the basal part of the explant in contact with the growth medium [88]. (**O**) Magnified clusters of vascular elements in the pits of treated explants, see also Figure (**M**) [88]. (**G**–**O**) Micrographs by light microscopy.

**Figure 7 ijms-25-05386-f007:**
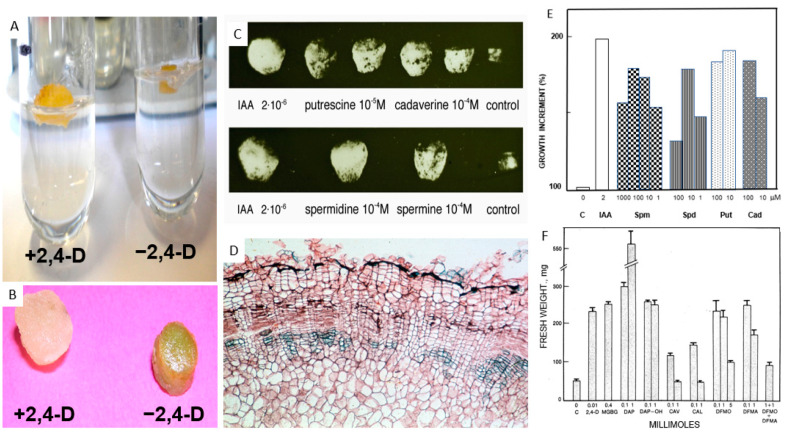
Effect of PAs and various related substances on the *Helianthus t.* long term-grown explant. (**A**) Comparison after 47 days of culture of the test tubes containing the explants treated with 2,4-D or untreated. (**B**) The same explants taken from the test tubes. (**C**) The explants after 40 days culture treated with IAA or Put, Cad, Spd and Spm and untreated (control). (**D**) Transverse section of an explant grown for 47 days in 0.1 mM Spm-containing medium showing continuous concentric rings of phloem and xylem, namely appressed vascular bundles [88]. (**E**) Percentage of growth increments of explants grown for 40 days in medium containing 2 mM IAA or different concentrations of PAs [53]. (**F**) The fresh weight (mg) of explants treated for 20 days of culture with a medium containing 2,4-D, or with 2,4-D added with MGBG (methylglyoxal bis-guanylhydrazone); DAP (diaminopropane), DAP-OH (hydroxy diaminopropane), CAV (canavanine), CAL (canaline), DFMO (alfa-difluoromethylornithine), DFMA (alfa-difluoromethylarginine), DFMO + DFMA [96].

**Table 1 ijms-25-05386-t001:** Biochemical features: weight, nucleic acids, proteins, PA biosynthesis, free and bound PAs during various stages of *Helianthus t.* L. tuber formation, dormancy and sprouting (PCD).

	*Tuber stages*	Tuber Formation			Tuber Dormancy			Tuber Sprouting	
							PCD	PCD	
	** *Tuber phases* **	initial	middle	final	initial	middle	final	initial	final
	** *Month* **	Mid-Sept---> Oct ------> Mid-Nov ------>Dec-Jan -------> Mid-Feb -----------> May							
	** *Biochemical features* **								
** *1* **	**Fresh Weight**	incr.	stat-incr.	incr	**max**	stat	stat	stat	stat
** *2* **	**rRNA**	decr	incr	incr	**max+**	**max+**	stat	decr	decr
** *3* **	**Total Proteins**	incr	stat/incr	stat/incr	stat	**stat/max**	**stat/max**	decr	decr
** *4* **	**38 kDa protein**	stat	incr	incr	incr	**max**	stat	decr	*min*−
** *5* **	**177 kDa protein**	no	no	no	no	**max**	decr	no	no
** *6* **	**Spm**	stat	**max**	decr	stat	incr	incr/stat	incr	**max+**
** *7* **	**Spd**	stat	**max**	decr	stat	stat	incr	incr	**max+**
** *8* **	**Put**	stat	stat	incr	stat	incr	incr/stat	incr	stat
** *9* **	**ODC**	stat	stat	stat					
** *10* **	**ADC**	incr	incr	**max**					
** *11* **	**SAMDC**	stat	incr	**max**					
** *12* **	**DAO**	decr	stat/incr	**max+**					
** *13* **	**Glutamine**				incr	incr	**max**	*min*	*min*−
** *14* **	**Arginine**				incr	incr	**max**	*min*	*min*−
** *15* **	**Bound Spm %**				incr	stat	stat	*min*	incr
** *16* **	**Bound Spd %**				decr	stat	stat	*min*−	incr
** *17* **	**Bound Put %**				incr	stat	decr	*min*−	incr
** *18* **	**Spm Bound to rRNA**				*min*	**max+**	*min*		
** *19* **	**Spd Bound to rRNA**				*min*	**max**	stat		
** *20* **	**Put Bound to rRNA**				*min*	**max**	*min*		

incr = increment; decr = decrement; stat = stationary; max = maximum; min = minimum; + = very high; − = very low; wide = not detected; no = absent.

## Data Availability

The data presented in this study are available on request from the corresponding author.

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
