# Peer review of "Programmed Cell Death Reversal: Polyamines, Effectors of the U-Turn from the Program of Death in Helianthus tuberosus L."

_ijms, 2024, doi:10.3390/ijms25105386_

Round 1

Reviewer 1 Report

Comments and Suggestions for Authors 1. This is a review paper, providing an overview of existing knowledge on programmed cell death in Helianthus tuberosus in connection to polyamine metabolism. 2. This paper is a review paper and provides an overview of materials published previously, no original data are reported in it and therefore no specific gap in the field is directly addressed. 3. It adds a historical review of information about the programmed cell death in Helianthus tuberosus. 4. Since it is a review paper, no controls and methods were described in it and there is no methods section. 5. It is a review paper providing a historical review of existing/published data, no experiments were conducted in the scope of this publication. All conclusions were made in respective publication that were cited and should be cited on the paper. Authors should improve citation of all references of the relevant research in this manuscript to address all questions on the topic. 6. Yes, but not all relevant references were included into the manuscript and should be updated. 7. Figures must be revised and include a clear rationale avoiding mixing up different data that are not linked with each other. Please see the review report for details.

Comments on the Quality of English Language

Moderate English editing, proof reading required

Author Response

Dear Reviewers,

we thank you for your effort in reviewing the manuscript; we have considered your criticisms which have been followed to modify and improve the manuscript. Parts that have been changed throughout the text are highlighted in red so you can find them quickly. Below is the detailed response to the criticisms you raised. Figs 3, 4, 5 and 6 have been modified

REV 1

REV 1: This is an interesting attempt to review existing knowledge on programmed cell death in Helianthus tuberosus in connection to polyamine metabolism. However, this paper has to be re-written with clear citations of relevant literature and improved scientific logic. Below are examples where to improve, but the whole manuscript must be optimized same way  since pribleslms addressed in following examples are repetitive in the full text.

REPLY: The paper had been changed according to referee criticism; clear citation of relevant literature has been added, moreover the article has been reorganized to improve the scientific logic. Schematically, the paper has been structured as follows:

  1. Introduction,
  2. The Helianthus tuberosus experimental system
  3. Main data from the research on the entire life cycle of Helianthus tuberosus

This part 3 consists of paragraphs that deal with:

The Tuber formation, Tuber Dormancy, Tuber sprouting and PCD as Natural Destiny. (Death Program) as well as the Break of tuber dormancy and in vitro cultures, indeed the Destiny change.  (Life Program).  The change of destiny towards a Life Program has been observed also in short-term experiments by Cultures untreated or treated with PAs or growth factors in which dormant cells break dormancy entering in a synchronous amitotic cell cycle of which the morphological, physiological cell cycle phases characteristics were highlighted. The response to PAs or growth factors, promotors, or inhibitors has been reported also for long-term experiments by culturing parenchyma cylinders and also in this case the morphological, physiological characteristics were highlighted. After a paragraph on cultures of Helianthus tuberosus experimental system as model to test drugs, PA and related antiproliferative compounds for health applicative purposes

  1. Conclusive Remarks, this final part highlight and discuss in the light of recent literature that it is possible to revert the Helianthus t. PCD, in the nature irreversible, to a life cell program as well as the multiple roles of aliphatic polyamines, which are described in the appendix part, independently from the main subject of the Manuscript, namely PCD. We believe that this is a schematic and logic way to present synthetically a long time period of research.

The appendix.  This appendix, that was suggested to be too long has been shortened considerably. The structure of this part remained as it was, but sentences or single words have been cancelled along the entire text. Moreover, a short history of the development of PA research has been added, according the comment of the referee, and a rich citation of references was included. As a consequence, the entire list of the cited literature has been revised.   In addition, some figures which appear too detailed for an appendix to be shortened, have been cancelled (in the present version Appendix has 3 Figures instead of 5 of the previous version and Fig 2 and 3 have been redrawn).

REV 1: Page 1 - "This review has also the aim to present old, neglected data, as many new reviews fail to mention papers published before or around the years 1970-90 not only on dormancy... "

Suggest to replace this general Information by a sequence of references, e. g. "this review

covers data reported on [x, y, z,...].

REV 1: Page 2, section 1.2. Development and PCD - this is action lacks references, first block of text does not contain citations of relevant literature

Reply: Thank you for suggestion; this section has been restructured and re-organized; according to Ref 1 suggestion many citations have been added as follows:

  1. Van Doorn, W.G.; Beers, E.P.; Dangl, J.L.; Franklin-Tong, V.E.; Gallois, P.; Hara-Nishimura, I.; Jones, A.M.; Kawai-Yamada, M.; Lam, E.; Mundy, J.; et al. Morphological Classification of Plant Cell Deaths. Cell Death Differ. 2011, 18, 1241–1246, doi:10.1038/cdd.2011.36.
  2. Jiang, C.; Wang, J.; Leng, H.-N.; Wang, X.; Liu, Y.; Lu, H.; Lu, M.-Z.; Zhang, J. Transcriptional Regulation and Signaling of Developmental Programmed Cell Death in Plants. Front. Plant Sci. 2021, 12, doi:10.3389/fpls.2021.702928.
  3. Rowarth, N.M.; Tattrie, S.B.; Dauphinee, A.N.; Lacroix, C.R.; Gunawardena, A.H.L.A.N. Filling in the Gaps: A Road Map to Establish a Model System to Study Developmental Programmed Cell Death. Botany 2023, 101, 301–317, doi:10.1139/cjb-2022-0110.
  4. Parveen, S.; Altaf, F.; Farooq, S.; Lone, M.L.; Ul Haq, A.; Tahir, I. The Swansong of Petal Cell Death: Insights into the Mechanism and Regulation of Ethylene-Mediated Flower Senescence. J. Exp. Bot. 2023, 74, 3961–3974, doi:10.1093/jxb/erad217.
  5. Rubinstein, B. Regulation of Cell Death in Flower Petals. Plant Mol. Biol. 2000, 44, 303–318, doi:10.1023/A:1026540524990.
  6. Serafini-Fracassini, D.; Del Duca, S.; Monti, F.; Poli, F.; Sacchetti, G.; Bregoli, A.M.; Biondi, S.; Della Mea, M. Transglutaminase Activity during Senescence and Programmed Cell Death in the Corolla of Tobacco (Nicotiana Tabacum) Flowers. Cell Death Differ. 2002, 9, 309–321, doi:10.1038/sj.cdd.4400954.
  7. Della Mea, M.; De Filippis, F.; Genovesi, V.; Fracassini, D.S.; Del Duca, S. The Acropetal Wave of Developmental Cell Death of Tobacco Corolla Is Preceded by Activation of Transglutaminase in Different Cell Compartments. Plant Physiol. 2007, 144, 1211–1222, doi:10.1104/pp.106.092072.
  8. Della Mea, M.; Serafini-Fracassini, D.; Del Duca, S. Programmed Cell Death: Similarities and Differences in Animals and Plants. A Flower Paradigm. Amino Acids 2007, 33, 395–404, doi:10.1007/s00726-007-0530-3.
  9. Cai, G.; Della Mea, M.; Faleri, C.; Fattorini, L.; Aloisi, I.; Serafini-Fracassini, D.; Del Duca, S. Spermine Either Delays or Promotes Cell Death in Nicotiana Tabacum L. Corolla Depending on the Floral Developmental Stage and Affects the Distribution of Transglutaminase. Plant Sci. 2015, 241, 11–22, doi:10.1016/j.plantsci.2015.09.023.
  10. Gentile, A.; Antognoni, F.; Iorio, R.A.; Distefano, G.; Casas, G.L.; La Malfa, S.; Serafini-Fracassini, D.; Del Duca, S. Polyamines and Transglutaminase Activity Are Involved in Compatible and Self-Incompatible Pollination of Citrus grandis. Amino Acids 2012, 42, 1025–1035, doi:10.1007/s00726-011-1017-9.
  11. Mandrone, M.; Antognoni, F.; Aloisi, I.; Potente, G.; Poli, F.; Cai, G.; Faleri, C.; Parrotta, L.; Del Duca, S. Compatible and Incompatible Pollen-Styles Interaction in Pyrus communis l. Show Different Transglutaminase Features, Polyamine Pattern and Metabolomics Profiles. Front. Plant Sci. 2019, 10, doi:10.3389/fpls.2019.00741.
  12. Serafini-Fracassini, D.; Di Sandro, A.; Del Duca, S. Spermine Delays Leaf Senescence in Lactuca Sativa and Prevents the Decay of Chloroplast Photosystems. Plant Physiol. Biochem. 2010, 48, 602–611, doi:10.1016/j.plaphy.2010.03.005.
  13. Del Duca, S.; Betti, L.; Trebbi, G.; Serafini-Fracassini, D.; Torrigiani, P. Transglutaminase Activity Changes during the Hypersensitive Reaction, a Typical Defense Response of Tobacco NN Plants to TMV. Physiol. Plant. 2007, 131, 241–250, doi:10.1111/j.1399-3054.2007.00950.x.
  14. Tiburcio, A.F.; Altabella, T.; Bitrián, M.; Alcázar, R. The Roles of Polyamines during the Lifespan of Plants: From Development to Stress. Planta 2014, 240, 1–18, doi:10.1007/s00425-014-2055-9.
  15. Del Duca, S.; Serafini-Fracassini, D.; Cai, G. Senescence and Programmed Cell Death in Plants: Polyamine Action Mediated by Transglutaminase. Front. Plant Sci. 2014, 5, doi:10.3389/fpls.2014.00120.
  16. Evans, P.T.; Malmberg, R.L. Do Polyamines Have Roles in Plant Development? Annu. Rev. Plant Physiol. Plant Mol. Biol. 1989, 40, 235–269, doi:10.1146/annurev.pp.40.060189.001315.
  17. Bagni, N.; Tassoni, A. Biosynthesis, Oxidation and Conjugation of Aliphatic Polyamines in Higher Plants. Amino Acids 2001, 20, 301–317, doi:10.1007/s007260170046.
  18. Moschou, P.N.; Roubelakis-Angelakis, K.A. Polyamines and Programmed Cell Death. J. Exp. Bot. 2014, 65, 1285–1296, doi:10.1093/jxb/ert373.
  19. Cai, G.; Sobieszczuk-Nowicka, E.; Aloisi, I.; Fattorini, L.; Serafini-Fracassini, D.; Del Duca, S. Polyamines Are Common Players in Different Facets of Plant Programmed Cell Death. Amino Acids 2015, 47, 27–44, doi:10.1007/s00726-014-1865-1.
  20. Kanamori, Y.; Finotti, A.; Di Magno, L.; Canettieri, G.; Tahara, T.; Timeus, F.; Greco, A.; Tirassa, P.; Gasparello, J.; Fino, P.; et al. Enzymatic Spermine Metabolites Induce Apoptosis Associated with Increase of P53, Caspase-3 and Mir-34a in Both Neuroblastoma Cells, SJNKP and the N-Myc-Amplified Form IMR5. Cells 2021, 10, doi:10.3390/cells10081950.
  21. Del Duca, S.; Serafini-Fracassini, D.; Cai, G. Senescence and Programmed Cell Death in Plants: Polyamine Action Mediated by Transglutaminase. Front. Plant Sci. 2014, 5, doi:10.3389/fpls.2014.00120.
  22. Bertossi, F.; Bagni, N.; Moruzzi, G.; Caldarera, C.M. Spermine as a New Growth-Promoting Substance for Helianthus Tuberosus (Jerusalem Artichoke) in Vitro. Experientia 1965, 21, 80–81, doi:10.1007/BF02144752.

REV 1: Page 2, section 2. -"The Plant model " this title does not specify and reflects the information that follows, please pick a better title of this large section

REPLY: The title has been changed with “The Helianthus tuberosus experimental system

"Helianthus t." -> H. tuberosus (in italics)

REPLY: “Helianthus t."  are now in Italics.      

REV 1: "Their multiple names show that they are well known in the world. The name Jerusalem artichoke has nothing to do with Jerusalem but derives from the incorrect pronounce from the Italian immigrants in the USA of the term “girasole” (sunflower), having a similar flower. “Artichoke” is due to the similarity of their taste with the former. " - this information has

nothing to do with the title and abstract of this paper …

REPLY: Thanks to Ref 1, but in this case, we think the sentence could remain in the text. The reason lies in the fact that since the article talks extensively about Heliathus tuberosus, from an etymological point of view it is interesting to know the names in common use in different countries to indicate the same plant.

REV 1: "The plant of Helianthus t. cv OB1 utilized is a kind gift of Prof. Roger J. Gautheret, who selected it and pointed out already in 1953 at Paris University the technique of in vitro culture of this line for its peculiar properties, being very stable and not developing roots in culture, as described below. This line was vegetatively reproduced and cultivated since 1964 in the Botanical Garden of the Department of Biological, Geological and Environmental Sciences (University of Bologna), at present directed by Dr Umberto Mossetti; in the 1980s, as a scholarship holder, he published some articles on tuber dormancy and in vitro culture, obtaining the first data on bound PAs (see Literature). " -

please specify the relevant literature after each statement, following journals formate

REPLY: Change (see Literature) with [22, 23, 24].

  1. Serafini-Fracassini, D.; Mossetti, U. Free and Bound Polyamines in Different Physiological Stages of Helianthus tuberosus Tuber. In Recent Progress in Polyamine Research; Selmeci L., Brosnan M.E., Seiler N, 1985; pp. 551–560.
  2. Mossetti, U.; Serafini-Fracassini, D.; Del Duca, S. Protein-Bound Polyamines in Helianthus tuberosus. Giornale Botanico Italiano 1986, 120, 89–89, doi:10.1080/11263508609429342.
  3. Mossetti, U.; Serafini Fracassini, D.; Del Duca, S.; D’Orazi, D. Conjugated polyamines during dormancy and activation of tuber of Jerusalem artichoke. In Proceedings of the Proceedings of the International Symposium Conjugated Plant Hormones, Structure, Metabolism and Function; K. Schreiber, H.R. Schutte, G. Sembnder: Berlino, 1987; Vol. 1, pp. 369–375

REV 1: Page 3 - Tab. 1 shows stages of development, but the cited Figure 1 does not, it shows different data not linked each other (e. g. microscopy of tissues combined with plant fotos)

Reply: The text has been changed according to referee’s suggestion, as follows: Table 1 shows the stages of development of the plant during the year. The plant stem with flower buds (Fig. 1 A), the developed dormant tubers showing the primordia of vegetative buds (Fig. 1 B) and the mature flower (Fig 1 C) are shown together with their tissue and cell details at TEM, as well as some aspects of cell death. In the revised version, Figure 1 and Table 1 has been separated.

REV 1: Page 3 - "Yang et al., 2015; Judprasong et al., 2018; Shariati et al., 2021" - change to format which is being used in the whole article - [x, y, z], there are different citation formattings in the same sentence

REPLY: change the sentence as follows: As examples, see reviews in the literature [26-28]

REV 1: Page 4 - "thus, the literature references are necessarily restricted mainly to some of our articles and, when possible, completed with other recent data." - which references? Specify

REPLY: the literature references are necessarily restricted mainly to some of our articles and completed with more recent data from literature reported along the text.

We believe that it is not appropriate to report here a mere list of references which are instead already reported promptly throughout the text;  

REV 1: "The research was focused on PAs, at that time promising growth factors only in humans and bacteria, and on their interaction with fundamental components of the cell, like nucleic acids. The initial target of this research was to verify if PAs were possible effectors of the program change acting simply as growth factors, but later also exploring their additional possible roles." - literature reference is missing.

REPLY: Thank you, literature have been added

Bertossi, F.; Bagni, N.; Moruzzi, G.; Caldarera, C.M. Spermine as a New Growth-Promoting Substance for Helianthus Tuberosus (Jerusalem Artichoke) in Vitro. Experientia 1965, 21, 80–81, doi:10.1007/BF02144752.

  1. Evans, P.T.; Malmberg, R.L. Do Polyamines Have Roles in Plant Development? Annu. Rev. Plant Physiol. Plant Mol. Biol. 1989, 40, 235–269, doi:10.1146/annurev.pp.40.060189.001315. "

Bachrach, U. The Early History of Polyamine Research. Plant Physiology and Biochemistry 2010, 48, 490–495, doi:10.1016/j.plaphy.2010.02.003.

Morgan, D.M.L. Polyamines: An Overview. Applied Biochemistry and Biotechnology - Part B Molecular Biotechnology 1999, 11, 229–250, doi:10.1007/BF02788682

Liquori, A.M.; Costantino, L.; Crescenzi, V.; Elia, V.; Giglio, E.; Puliti, R.; De Santis Savino, M.; Vitagliano, V. Complexes between DNA and Polyamines: A Molecular Model. Journal of Molecular Biology 1967, 24, 113–122, doi:10.1016/0022-2836(67)90094-0.

Quigley, G.J.; Teeter, M.M.; Rich, A. Structural Analysis of Spermine and Magnesium Ion Binding to Yeast Phenylalanine Transfer RNA. Proceedings of the National Academy of Sciences of the United States of America 1978, 75, 64–68, doi:10.1073/pnas.75.1.64.

Feuerstein, B.G.; Pattabiraman, N.; Marton, L.J. Molecular Mechanics of the Interactions of Spermine with DNA: DNA Bending as a Result of Ligand Binding. Nucleic Acids Research 1990, 18, 1271–1282, doi:10.1093/nar/18.5.1271.

Iacomino, G.; Picariello, G.; D’Agostino, L. DNA and Nuclear Aggregates of Polyamines. Biochimica et Biophysica Acta - Molecular Cell Research 2012, 1823, 1745–1755, doi:10.1016/j.bbamcr.2012.05.033).

REV 1: Page 5 -"Results" - why there is a results section in a review paper? "The most significant biochemical features, even though partial, due to the complexity of the tuber metabolism, are summarized in Table 2." - what does partial mean, are these data

not complete? What is a source of this data (for a review paper reference is needed)

REPLY: The title “Results” has been changed with: “Main data from the research on the entire life cycle of Helianthus tuberosus

The sentence “even though partial, due to the complexity of the tuber metabolism,” has been removed, the revised sentence is:

“The most significant biochemical features, are summarized in Table 2.”

For each of the parameters reported in Table 2, the appropriate literature has been cited and reported under the Table 2

REV 1: Page 5-18 - please check all reference citations for mentioned data and link figures to information mentioned in text in different sections

REPLY:  References have been checked and some parts of the text eliminated (as for ex the sentence in 3.2. Tuber Formation “When the cells are no more dividing (when only ODC was active), but are simply expanding during middle formation phase, only ADC is involved, whereas ODC activity is substantially inactive. These observations shed light on the debated question of which of the two biosynthetic pathways is activated according to a particular cell physiological condition. These data clearly show that …” and others)

REV 1: Page 18 - Discussion section begins with conclusions, is not this section "Conclusions"?

REV 1: Please update references in the last section

REPLY: References have been updated

Reviewer 2 Report

Comments and Suggestions for Authors

The author appears to have attempted to compile a comprehensive history of Helianthus tuberosus L. However, we are concerned that the quality of the data used in the figures is clearly low compared to these efforts. Although it is good to use existing data in figures, it is important to thoroughly review and verify the developed data. A more appropriate approach would be to write a historical review of Helianthus tuberosus L., improved and newly created to the high standards of this journal.

Simply increasing the volume of reviews may not make a positive impression on researchers who read your journal. Instead, we recommend tabulating the history of Helianthus tuberosus L. research, accompanied by newly created figures and diagrams that adhere to strict quality standards.

I'm sorry to the authors who wrote a lot of content, but the content should be written in a more organized manner. Also, the content included in the supporting infomration also contains too much information.

Author Response

Dear Reviewer,

we thank you for your effort in reviewing the manuscript; we have considered your criticisms which have been followed to modify and improve the manuscript. Parts that have been changed throughout the text are highlighted in red so you can find them quickly. Below is the detailed response to the criticisms you raised. Figs 3, 4, 5 and 6 have been modified

REV 2

The author appears to have attempted to compile a comprehensive history of Helianthus tuberosus L. However, we are concerned that the quality of the data used in the figures is clearly low compared to these efforts. Although it is good to use existing data in figures, it is important to thoroughly review and verify the developed data. A more appropriate approach would be to write a historical review of Helianthus tuberosus L., improved and newly created to the high standards of this journal.

REPLY: According to referee’s criticism the text of the manuscript has been definitely reorganized and  figures 3, 4, 5, 6, of the main manuscript have been revised and modified.  The same has been done for the Appendix where the figures have been reduced to three and simplified, eliminating the original graphics or models of historical interest taken from the original papers (even though well recognized and accepted by the specialised scientific community, thus more than verified), now presenting simple schemes.

A short historical review has been newly created on PAs, citing the studies performed along the development of the research in Helianthus and in other plants. The historical review of Helianthus t has been already described in detail in the main manuscript of this paper and this is a completely new work (see below).  I would only underline that the relative citations were chosen from the probably more than 200 peer reviewed papers that were published by our lab. Thus, to summarize all these data, is not simple and easy to "tabulate" them.  To certificate the recognised seriousness of our data, which seems necessary to be verified, I, DSF, have been recognized as one of the only seven scientists, among which U.Bachrach; J-P Moulinoux, O. Heby, lo Persson, C. Bacchi, l. Rusty of the Honor Committee of the International Polyamines Foundation, and I am also certificated reviewer of MDPI.

REV 2: we recommend tabulating the history of Helianthus tuberosus L. research, accompanied by newly created figures and diagrams that adhere to strict quality standards.

REPLY: According to the comment of the referee, in the appendix section a short history of the development of PA research in Helianthus tuberosus L.  has been added and a rich citation of references was included. In addition, some figures which appear too detailed for an appendix to be shortened (to this concerns, Referee wrote “ Also, the content included in the supporting infomration also contains too much information”), have been cancelled. Two of the previous 5 figs (now Fig 2 and 3) have been redrawn, the present version of the appendix contain 3 figures instead of 5.

REV 2: the content should be written in a more organized manner

REPLY: According to Referee suggestion, the article has been reorganized to improve the scientific logic. Schematically, the paper has been structured as follows:

  1. Introduction,
  2. The Helianthus tuberosus experimental system
  3. Main data from the research on the entire life cycle of Helianthus tuberosus

This part 3 consists of paragraphs that deal with:

The Tuber formation, Tuber Dormancy, Tuber sprouting and PCD as Natural Destiny. (Death Program) as well as the Break of tuber dormancy and in vitro cultures, indeed the Destiny change.  (Life Program).  The change of destiny towards a Life Program has been observed also in short-term experiments by Cultures untreated or treated with PAs or growth factors in which dormant cells break dormancy entering in a synchronous amitotic cell cycle of which the morphological, physiological cell cycle phases characteristics were highlighted. The response to PAs or growth factors, promotors, or inhibitors has been reported also for long-term experiments by culturing parenchyma cylinders and also in this case the morphological, physiological characteristics were highlighted. After a paragraph on cultures of Helianthus tuberosus experimental system as model to test drugs, PA and related antiproliferative compounds for health applicative purposes

  1. Conclusive Remarks, this final part highlight and discuss in the light of recent literature that it is possible to revert the Helianthus t. PCD, in the nature irreversible, to a life cell program as well as the multiple roles of aliphatic polyamines, which are described in the appendix part, independently from the main subject of the Manuscript, namely PCD. We believe that this is a schematic and logic way to present synthetically a long time period of research.

We hope the revise version of the manuscript match with the standard of the IJMS

Best regards,

Stefano Del Duca & Donatella Serafini-Fracassini

Reviewer 3 Report

Comments and Suggestions for Authors

This manuscript provides some comprehensive summary of programmed cell death reversal special emphasized on polyamines.

The authors provided many new information and readers will be benefited from this synthesis.

However, I suggest the authors following the standard format of a review article. For example, they provided a section "Discussion". I suggested separating them into relevant sections and furnish the literature.

Make the content simple and easier.

Figure quality could be improved.

Some of the references are old.Please update them.

Author Response

Dear Reviewers,

we thank you for your effort in reviewing the manuscript and thank you for your comments. We have considered your criticisms which have been followed to modify and improve the manuscript. Parts that have been changed throughout the text are highlighted in red so you can find them quickly. The quality of figure has been improved by modifying Figs 3, 4, 5 and 6; the text has been also modified and Discussion has been changed with Conclusive remarks, and also its content has been changed.

Best regards,

Stefano Del Duca

Donatella Serafini-Fracassini
